# COVID-19 Vaccinations, Trust, and Vaccination Decisions within the Refugee Community of Calgary, Canada

**DOI:** 10.3390/vaccines12020177

**Published:** 2024-02-09

**Authors:** Fariba Aghajafari, Laurent Wall, Amanda Weightman, Alyssa Ness, Deidre Lake, Krishna Anupindi, Gayatri Moorthi, Bryan Kuk, Maria Santana, Annalee Coakley

**Affiliations:** 1Department of Family Medicine, Cumming School of Medicine, University of Calgary, Calgary, AB T2N 1N4, Canada; 2Department of Community Health Sciences, Cumming School of Medicine, University of Calgary, Calgary, AB T2N 1N4, Canada; 3Cumming School of Medicine, University of Calgary, Calgary, AB T2N 1N4, Canada; 4Habitus Consulting Collective, Calgary, AB T2T 1P3, Canada; laurentwall@habituscollective.ca (L.W.); amandaweightman@habituscollective.ca (A.W.);; 5Alberta International Medical Graduates Association, Calgary, AB T2E 3K8, Canada; 6Mosaic Refugee Health Clinic, Calgary, AB T2A 5H5, Canada

**Keywords:** COVID-19, undervaccination factors, vaccine hesitancy, vaccination barriers, vaccine confidence, vaccine uptake, trust

## Abstract

Refugee decisions to vaccinate for COVID-19 are a complex interplay of factors which include individual perceptions, access barriers, trust, and COVID-19 specific factors, which contribute to lower vaccine uptake. To address this, the WHO calls for localized solutions to increase COVID-19 vaccine uptake for refugees and evidence to inform future vaccination efforts. However, limited evidence engages directly with refugees about their experiences with COVID-19 vaccinations. To address this gap, researchers conducted qualitative interviews (N = 61) with refugees (n = 45), sponsors of refugees (n = 3), and key informants (n = 13) connected to local COVID-19 vaccination efforts for refugees in Calgary. Thematic analysis was conducted to synthesize themes related to vaccine perspectives, vaccination experiences, and patient intersections with policies and systems. Findings reveal that refugees benefit from ample services that are delivered at various stages, that are not solely related to vaccinations, and which create multiple positive touch points with health and immigration systems. This builds trust and vaccine confidence and promotes COVID-19 vaccine uptake. Despite multiple factors affecting vaccination decisions, a key reason for vaccination was timely and credible information delivered through trusted intermediaries and in an environment that addressed refugee needs and concerns. As refugees placed trust and relationships at the core of decision-making and vaccination, it is recommended that healthcare systems work through trust and relationships to reach refugees. This can be targeted through culturally responsive healthcare delivery that meets patients where they are, including barrier reduction measures such as translation and on-site vaccinations, and educational and outreach partnerships with private groups, community organizations and leaders.

## 1. Introduction

Research shows that in many high-income countries, including Canada, recent migrants, refugees, and asylum seekers are at an increased risk of being undervaccinated [1]. In the context of the COVID-19 pandemic, early research on COVID-19 vaccine coverage found that immigrants and refugees were disproportionately impacted by the COVID-19 pandemic, yet their vaccine uptake was lower [2] and hesitancy was reported to be higher than in the general population [3]. This study builds on COVID-19 research aimed at addressing factors related to lower rates of vaccination for refugees by examining how individual patient experiences can contribute to more effective policies and systems.

Vaccine hesitancy is defined by the World Health Organization as the “delay in acceptance or refusal of vaccination despite availability of vaccination services” [4]. Given the broad scope of hesitancy and its application to patients, they argue that a thorough understanding of the context and access to healthcare and vaccination services is required prior to classifying patients as vaccine hesitant, as marginalized communities have health issues that are linked to medical distrust and structural racism [4].

A closely related term to vaccine hesitancy is vaccine confidence, which covers a range of concepts, including trust in vaccines, views on vaccine safety, trust in healthcare workers delivering the vaccine, and trust in the vaccine approval process [5]. Given these related factors, determinants of undervaccination is a more holistic term that captures the myriad of intersecting individual, community, structural, legal, and technical factors which can impede access or acceptance of vaccines. These include information inequities [3,4,6,7,8,9,10,11], personal beliefs [1,2,6,7,8,12], previous systems experiences [1,4,6,7,9,11,13,14] structural inadequacies [1,7,13,14], accessibility of vaccine services [1,5,7,13], and intersecting risk factors for vaccinations [1,6,7,8,12,13,14,15] (see Appendix A.

Evidence to address vaccine preventable diseases emphasizes the importance of robust health systems that prioritize high risk and excluded groups for vaccine delivery [1,16]. Rather than a one-size-fits-all strategy toward vaccination, standards and approaches should be adapted to diverse contexts [7] and be attentive to diversity within migrant populations [16]. Immigrant populations, due to various intersecting factors, remain largely hesitant and more at risk compared to general populations [1]. In this context, recommended strategies include tailored, culturally competent, and migrant sensitive approaches based on an understanding of community-specific barriers, beliefs, practices, and facilitators [7,16,17,18]. Furthermore, understanding hesitancy within specific vulnerable populations is critical for supporting uptake [19]. Evidence on vaccine confidence highlights the importance of interpersonal and community factors in shaping individual views and vaccine uptake [5,20]. For example, trust and confidence are built within a context of relationships between different individuals and between individuals and public health systems. These relationships have been identified as key promoters [10,21] or barriers [22] to vaccination.

Non-medical vaccine research on immigrant and refugees has predominantly taken the form of cross-sectional surveys, longitudinal surveys, and secondary data analysis to study vaccination intent, hesitancy, and/or attitudes as well as relating factors in COVID-19 or other vaccine contexts (e.g., [23,24,25,26]). Other core areas of study have focused on models of vaccination and the delivery of vaccines (e.g., [27,28]). While the COVID-19 pandemic has led to an influx of new scholarship on these topics, there is a gap with regards to the experiences of patients within vaccination systems, especially marginalized patients.

An important contribution of early qualitative research on COVID-19 vaccine hesitancy was identifying that racialized residents who received first doses were “hesitant, rather than opposed” [19], which frames vaccine hesitancy as part of a decision-making process rather than a finite decision. Evidence highlights various experiences that help racialized and/or immigrant patients make informed decisions, as they draw on various touch points with the local networks, health systems, and global community when making decisions [19,29]. Key themes in the decision-making process for American racialized immigrants were a lack of sufficient and accurate information, a lack of answers to common questions about the virus, deeply rooted historical underpinnings for hesitancy (e.g., mistreatment by the medical community), and limited accessibility of vaccines [30]. These studies highlight the interplay of various decision-making factors, including ones related to undervaccination, and that a low proportion of racialized and marginalized persons are firmly against vaccination.

Limited qualitative scholarship on patient perspectives and vaccine decision-making processes has been conducted in the Canadian, and especially Albertan, context. One such example argued that while public health communication strategies existed in Alberta, the strategies did not factor in the demographic of immigrant mothers and target them appropriately [10]. Despite this, decisions by immigrant mothers were largely in favour of vaccination following interactions with healthcare personnel [10].

Gaps in scholarship also highlight that research needs to focus on vaccination decisions, including factors of fear and misinformation in the COVID-19 context, as well as the effects of targeted public health messaging on racialized patients, as it may be counterproductive in some contexts [31]. While trust, a desire to be meaningfully integrated into vaccination decisions, clear information strategies, the integration of marginalized perspectives into vaccination strategies, and trusted social circles are consistent factors that help address hesitancy, there is limited evidence on the perspectives, experiences, and decision-making factors of the most vulnerable individuals in a society, such as refugees. Additionally, no localized evidence exists on refugee patient perceptions within Calgary and area vaccination systems, and the interplay of factors related to decisions to vaccinate. This study took an exploratory qualitative approach to address these gaps with the purpose of explicitly informing policy and systems recommendations through patient perspectives and experiences.

## 2. Methods

### 2.1. Study Design

This research was led by the University of Calgary, Departments of Family Medicine and Community Health Sciences. The research team included members of the University of Calgary, a research-intensive university in Canada, members of Habitus Consulting Collective, a research and evaluation consultancy with extensive experience working with newcomers, and three partners embedded in Calgary area vaccine infrastructure for refugees. The purpose of the broad research team was to bring together diverse research skill sets and facilitate access to refugees and stakeholders embedded within COVID-19 vaccination systems. These partners included the Calgary Catholic Immigration Society (CCIS), The Alberta International Medical International Graduates Association (AIMGA), and the MOSAIC Refugee Health Clinic. CCIS is a large immigrant service providing agency based in Calgary that was contracted to work directly with refugees and support them with resettlement during 2021–2022. AIMGA is an association that focuses on the integration of internationally trained physicians in Canada. Through their efforts to re-engage internationally trained physicians into Canadian healthcare systems they became heavily involved in COVID-19 vaccination efforts in 2021–2022 due to high demand for their members, as refugees responded well to members’ medical knowledge, cultural knowledge, and diverse language skills. They operated in diverse capacities in vaccination efforts in 2021–2022. The Mosaic Refugee Health Clinic is a Calgary clinic that provides refugees with primary care for a period of up to two years following entry into Canada, with rich networks and knowledge of current practices.

Researchers took an exploratory approach to study refugee experiences in the Calgary area from Spring 2021 to Fall 2022. The goal was to focus on the perspectives of refugee patients to better understand localized vaccination systems and provide recommendations for future healthcare delivery. As a qualitative study focused on a unique group, this study was designed to respond to a need for qualitative data that helps align vaccination interventions to how people think, feel, and act in relation to COVID-19 vaccines [32]. Furthermore, it was designed to address issues of qualitative validity and reliability by drawing on multiple qualitative data sources to converge on key themes, using specific sampling techniques to deliberately include a wide range of participants with knowledge and experience about a specific social phenomenon, building in feedback loops from insiders in the field of refugee vaccinations through the inclusion of research partners, and by developing the data collection instruments and analytical frameworks as a group [33]. These are outlined in more detail in this section.

### 2.2. Sampling and Recruitment

The research team conducted a series of qualitative semi-structured interviews, structured interviews, and a group interview with refugees and key informants connected to various local COVID-19 vaccination efforts for newcomers and refugees in Calgary, Alberta. Data collection was made possible through a research partnership with CCIS—the Resettlement Assistance Provider (RAP) for Southern Alberta (meaning the contract holder for settling all Government Assisted Refugees), AIMGA and the Mosaic Refugee Health Clinic. This project was approved for ethics by the University of Calgary Conjoint Health Research Ethics Board (CHREB REB21-0859).

A total of 61 participants were included in data collection. The majority of participants were healthcare system patients. They included government-sponsored refugees from Afghanistan who landed in Calgary in 2022 and were processed through local immigration systems and received COVID-19 vaccinations at processing hotels. Others were Private Sponsors of Refugees who helped refugees access COVID-19 vaccines, and longer-term refugees who had accessed vaccinations prior to the arrival of Afghan refugees.

The project included a series of brief, structured interviews with newly arrived Government Assisted Refugees (GARs) from Afghanistan (N = 39) at processing hotels and a series of longer, semi-structured interviews with Privately Sponsored Refugees (PSRs) (N = 6) and Private Sponsors of Refugees (N = 3). Additional semi-structured interviews (N = 11) and a semi-structured group interview (N = 2) were also completed with key informants, including community organization staff, vaccine access advocates, doctors, public health nurses and internationally trained medical graduates.

Primary data were collected from March 2022 to May 2023 with participants recruited through purposive sampling, convenience sampling, and snowball sampling. Afghan GARs were recruited by convenience sampling facilitated by CCIS and AIMGA immediately following their vaccination on-site at a temporary housing facility. Prior to vaccination, participants were offered a virtual information session in Dari and Pashto, introduced to the study purpose, given an opportunity to ask questions, and self-selected for participation. PSRs and Sponsors were recruited through purposive sampling by CCIS staff. These clients were given information about the study and self-selected for participation. Non-Afghan GARs were solicited but declined to participate. Key informants were identified by project partners who supported making connections as needed, as well as through snowball sampling. Consent for GAR interviews was verbal to build rapport through a paperless and less formal experience, and written for all other interviews. Consent forms, scripts, and interview guides for refugees were translated into first languages through certified translators and data collectors. All interview participant groups were provided a cash honorarium for their participation to reflect the participant’s expertise and time commitment; the exception was GAR participants, who had brief interviews.

### 2.3. Data Collection

Researchers explored the experiences of refugees and key informants with different types of interview guides for each participants group. Interviews with GARs, PSRs, and Sponsors focused on patient experiences with COVID-19 vaccinations, issues with vaccinations, concerns, and recommendations, with a semi-structured format being used for the latter two groups to ensure room for diverse avenues of conversation. The structured interview format for the former group, GARs, allowed researchers to conduct numerous interviews within specific and limited time frames and compare answers between respondents. Interviews with key informants were semi-structured and focused on descriptions of vaccination models, how they changed over time, strengths of models, barriers to vaccination, strategies to address barriers, trends with patients, and key lessons. As vaccination strategies by key informants differed over the course of the study, interviews with key informants adapted to focus on emergent findings, such as specific ways of encouraging vaccine uptake where details were sparse. The research team concluded that a point of data saturation was reached when no new patterns were identified through interviews.

The interviews were conducted in face-to-face, telephone, and video conference (Zoom) format, ranging between 5–45 min, after obtaining informed consent. Structured GAR interviews were face-to-face and ranged between 5 and 10 min. Semi-structured interviews were conducted over telephone for PSRs and Sponsors and over Zoom for key informants. These semi-structured interviews lasted from 20 to 45 min. Researchers conducted interviews with refugees in their preferred language. As this required numerous interviewers, a designated team member delivered interviewer training sessions to cover the research method, note taking, research topic, informed consent, working with refugees in a culturally sensitive manner, and transcription. All interviews were audio recorded and transcribed in English, with the exception of a few participants who preferred to not be recorded. In such cases, interviewers took in-depth notes during and following the session. The designated team member worked with first-language interviewers throughout the process of data collection to address questions and/or concerns, provide ongoing training, and ensure a consistent level of quality for translations and transcriptions.

### 2.4. Data Analysis

The research team conducted qualitative thematic analysis on interview data to ensure the analysis was grounded in components such as participant accounts and scholarship [34]. Throughout this process researchers drew on multiple frameworks of interpretation to make sense of the data, including researcher knowledge and experience, insights drawn from the data, and relevant scholarship and theory on COVID-19, vaccinations, and refugees. Given the need for a broad team of researchers to work together, researchers first piloted and refined a code guide to focus on research questions, and then used the code guide to code and organize interview data into initial categories and subcategories. Following this round of coding, sorted data was then used to develop initial themes and subthemes by adding nuances and insights to the initial categories. These themes then went through multiple rounds of refinement, which included writing, discussion, and routinely referring to original data, the research questions, and existing scholarship. The last step was to create final thematic write-ups with themes and subthemes, which focused on patient experiences with vaccinations, perceptions of vaccines, decision making factors, how hesitancy and confidence manifest themselves, and the interplay of factors related to decisions to vaccinate. Researchers also supported the coding and thematic analysis by reviewing interview notes, and discussing emergent findings with first language interviewers and the research team to ensure no themes were missed.

## 3. Results

Table 1 below outlines participant characteristics. The research team recruited diverse key informants to learn about refugee specific vaccination systems and gain insights into the experiences of refugees within systems. At the time of data collection, GARs accessible through CCIS were all from Afghanistan. Efforts were also made to speak to other groups of GARs; however, these efforts did not yield any participants. A small group of PSRs and Sponsors from diverse backgrounds were also recruited. Although researchers tried to systematically collect demographic information such as age range and country of origin, the research team was not able to reliably compile this information for all refugees. What was confirmed was that all 39 GARs originated from Afghanistan, some PSRs and Sponsors identified as being from Jordan or Ethiopia, and some did not specify. Key informants also reported working with population groups such as Arabs, West Africans, East Africans, South Asians, and Southeast Asians.

Refugee decisions to vaccinate for COVID-19 are a complex phenomenon that include vaccine hesitancy, vaccine confidence, and vaccine uptake, along with contextual and structural factors associated with the COVID-19 global pandemic. The experiences of refugees provided a rich picture of the context and motivations that surround decisions regarding vaccination. Given that refugees often face multiple barriers to health, their stories shed light on the complexities surrounding refugee health and have crucial implications for health systems.

Drawing on the social ecological model that understands health to be impacted by the interaction between the individual, the group/community, and the physical, social, and political environments [35], our findings identify the multi-level nature of impact, which is outlined in Figure 1. While this research could not determine how any one factor alone shaped vaccine uptake, this research highlights actionable recommendations in refugee healthcare delivery and contributes towards scholarship on vaccine hesitancy, vaccine confidence, and vaccine uptake.

Refugee and sponsor accounts focused predominantly on individual, interpersonal, and community factors. Key informants involved in vaccination systems, on the other hand, provided numerous community and structural factors. The following summary outlines the various factors involved in refugee decision-making. Participant responses linked to these factors are presented at the end of this section.

### 3.1. Individual Factors

Concerns about side effects: Concerns refugees had or held about the negative side effects of the vaccine and mortality rates were discussed by several refugees (n = 6) as a factor shaping their decisions. Concerns were for themselves or for their family members, and these sentiments tended to correspond with informal knowledge networks or broader community discourse. Such concerns were generally addressed when healthcare professionals took the time to ask about and address their concerns.Personal ‘disbelief’ in vaccine necessity or effectiveness and/or preference to avoid medical intervention: Another factor shared by refugees (n = 7) was a belief that the COVID-19 vaccine, or vaccines/medical intervention in general, is not effective or not a necessary form of defence against illness. This factor included the belief that their bodies are able to defend against diseases naturally and do not need additional assistance from vaccines.Concerns about risks to subpopulations: Individual refugees and all key informants involved in service provision spoke about vaccine safety concerns for specific subpopulations, such as children, youth, pregnant women, and persons with allergies or underlying health conditions. Similar to concerns about vaccine side effects, participants shared that concerns about specific subpopulations were addressed when healthcare professionals took the time to ask about and address such concerns.Fear of COVID-19: An important individual factor was the willingness and eagerness to be vaccinated. Refugees linked vaccine compliance to desires to mitigate physical health risks of COVID-19, including for healthy and at-risk persons. This motivation for health ultimately superseded any uncertainty about the vaccine or its safety, even in cases where refugees had remaining concerns, and key informants noted an ongoing desire to protect children with initial doses.

### 3.2. Interpersonal Factors

(Mis)information: Information was a key factor in shaping decisions which could positively or negatively influence vaccine uptake. Refugees shared examples of multiple and competing sources of information and linked these to feelings of hesitancy and confidence with the vaccine. Sources of misinformation commonly cited included social media and stories from friends and family. Despite being immersed in multiple sources of information, the large majority of refugees interviewed shared they were able to navigate misinformation and were not opposed to vaccinations. However, refugees also feared the influence of misinformation on other people who may be hesitant to vaccinate.Desire to protect others: Interpersonal factors included desires to protect others through vaccination. For example, some refugees framed their decision to be vaccinated as being for the protection for others, especially family members, rather than themselves.Influence of family members: Similar to (mis)information, a key factor that refugees contended with was the influence of family members. Refugees expressed different levels of hesitancy and/or vaccine confidence among family members, which shaped their thoughts and feelings around vaccinations. There were also cases where refugees explicitly stated that pressure or preference from family members, such as spouses, was a deciding factor in being vaccinated.

### 3.3. Community Factors

Information overload: While multiple and competing sources of information shaped vaccination decisions, according to participants, a key factor related to this was information overload. This factor had the potential to erode any positive gains in vaccine confidence that information from trusted and reliable sources could contribute to. For example, refugees linked the recurring changes in public health information to feelings of confusion, being overwhelmed, and uncertainty, especially with regards to information about the minutiae around risks and effectiveness of various brands of vaccines. While this wasn’t a factor in vaccine refusal per se, it added difficulty in determining personal vaccine confidence and participants highlighted desires to ‘shop around’ for specific vaccines.Access to evidence-based information: Evidence-based information was highlighted by refugees and key informants as key to gaining confidence. Access to such information was facilitated by multiple sources, including social media, primary sources, family members, and information sessions at vaccine clinics. While access could be facilitated through different sources, both refugees and key informants identified that evidence-based information helped increase confidence when it was delivered in a timely and trusted manner through personnel such as nurses or doctors prior to vaccination.Secondary information sources and personal networks: Another route to information was through secondary sources, such as social media, television, friends, and acquaintances. Refugees shared that these sources frequently delivered information related to the virus and the vaccine, whether they were actively seeking information (such as searching on the internet) or coming it across it more passively (such as hearsay). Family and friendship networks were also flagged as common sources of counsel and information about the vaccine.Pre-migration experiences: A more subtle factor in participant accounts was how refugees’ experience in Canada and their pre-migration experiences with vaccine access shaped their outlooks. This contributed to unique perceptions and concerns, such as whether patients perceived vaccination personnel to be open to questions, whether they had a choice in being vaccinated, or comparisons to other countries. Others noted that Canada had broad allowance for variations of personal preference in choosing the vaccine.Fatigue, indifference, and booster-specific hesitancy: The dose in question was a factor which influenced whether refugees pursued vaccinations or not. All participant groups shared cases where they, a family member, or a patient who conformed to or were even eager to get first and/or second doses refused subsequent boosters. Justifications for booster refusals varied, with examples of vaccine fatigue, indifference, or not wanting to go beyond the mandatory two doses. Key informants also reported that booster hesitancy trends followed the hesitancy patterns of early doses of the vaccine, with concerns around boosters’ side-effects and risks to subpopulations, with the added layer of fatigue and indifference.

### 3.4. Societal Factors

Table 2 below outlines the vaccination factors involved in decision-making that emerged at a societal level. Drawing on the social ecological model of health [35], these include broad societal factors that impact health, such as aggregates of social patterns which include language, cultural ideology, social institutions, and institutional practices, which contribute to health decisions and outcomes. Structural factors here refer specifically to factors that relate to social institutions and their practices, such as health policies.

Most of the recently arrived refugees did not mention these factors in interviews. Researchers ascribed this omission to refugees’ recent arrival, power differences between refugees and others in the host nation, and a lack of time to reflect on Canadian system experiences, and found that many barriers were addressed by the on-site vaccination program. While processing Afghan refugees did not disclose many of these factors in their accounts, Key Informants, Privately Sponsored Refugees, and Sponsors highlighted various examples in their accounts. See Table 3 for participant responses.

## 4. Discussion

This research focused on the experiences of refugees with COVID-19 vaccination systems within and around Calgary, Canada. The purpose was to understand vaccine hesitancy, barriers to access, decision-making factors, and refugee patient experiences during the COVID-19 vaccination roll out. This paper argues that vaccine confidence, hesitancy, uptake, and vaccination intent are not mutually exclusive. In fact, their complex relationship sheds light on how data on vaccine confidence and hesitancy need to be decoupled from data on vaccinations. Our paper shows that, for recently arrived refugees, the COVID-19 vaccination process became a testing ground for building and negotiating their relationship with a new healthcare system, and offers implications for strengthening the broader healthcare system’s capacity to serve refugee patients and implications for service providing collaborations focused on refugee patients.

The purpose of this study was to deepen the understanding of vaccine hesitancy, barriers to access, decision-making factors, and refugee patient experiences in the COVID-19 vaccination context. Our study found that, while processing refugees may not have had options to opt out of vaccination, as mandates shaped their capacity to travel, access amenities, or access employment, there were cases where refugees still had concerns about the vaccine itself despite receiving vaccination and information about the vaccine. Refugees also shared accounts of what was useful for their decision-making. To further complicate matters, vaccine rollouts over the 2021–2022 period included multiple doses (from first dose to boosters), multiple waves of eligibility, such as adults, seniors, and young children, and multiple access points over time. In fact, the study points to the complexities that shape decisions around COVID-19 vaccination and offers insight into how refugees and other marginalized groups come to engage and build a relationship with and through the health system.

Although some accounts were not specific to the COVID-19 vaccine, refugees framed their vaccine narrative in a broader context of health and migration, commonly citing issues such as lengthy wait times, multiple phone calls, complexity of systems, the constant need for paperwork in English, and receiving multiple doses due to previously unaccepted vaccines. On the other hand, many refugees were grateful for the fact that Canada had systems in place for COVID-19 vaccinations alongside supports for refugees. While the findings of this study were consistent with much of the global research understanding hesitancy with attention to context, intersectional identities, as well as possible barriers to access [5,7,19,29], findings have also provided insight into how these factors shape the overall relationship of refugees to the health system and ultimately health outcomes.

Vaccination personnel, including nurses, case managers, doctors, language support staff, and outreach staff, all played a part in shaping the experiences of patients. We provided evidence to support that relationships between refugees and vaccination system personnel were a key factor in encouraging uptake, even in the context of mandates. As such, their experiences place trust and relationships at the core of decision-making and vaccination. While refugee perspectives shared examples of touch points with local networks, health systems, and the global community [19,29] when making vaccination decisions, their experiences highlight that COVID-19 vaccination may have been less about vaccination outcomes than the process of vaccination. This is a crucial lesson for systems to incorporate when working with refugees as it places patient experiences and touch points at the heart of vaccination and supports research that the large majority of refugee patients were ready to engage in dialogue, “rather than opposed” to COVID-19 vaccination [19]. Similar to recent studies on racialized immigrants, only a low proportion of participants shared accounts that indicated a firm opposition to vaccination [10]. Stakeholder and refugee accounts also demonstrated that opposition to vaccination or reluctance to vaccinate grew for booster shots, and both groups shared accounts that refugee concerns and perspectives in this context mirrored broader trends in COVID-19. Researchers found no evidence that refugees felt negatively targeted by any of the measures taken by systems and personnel in the COVID-19 context.

In addition to centring patient experiences at the core of decision-making, this study has key findings. The first key finding was that vaccination uptake at any given time was context-dependent. Informants observed broad ebbs and flows in public demand and willingness to be vaccinated that corresponded with changing public policy and public discourse. A key theme from informants was that vaccine sentiments and vaccination rates were shaped by several environmental or contextual factors, and that staff activities had to monitor broader trends and change accordingly to ensure that refugees had accurate and relevant information. Importantly, the various touch points that patients drew on when making decisions were not the same for individual patients, yet these paralleled longer-term trends in perspectives towards COVID-19 vaccination. Despite this, services were able to target these touch points in culturally responsive manners through measures such as information sessions in first languages, and by routinely including personnel from similar cultural backgrounds alongside refugees.

Second, vaccine confidence and hesitancy were multifaceted and did not necessarily dictate vaccine uptake or a lack of uptake. Interviews provided insights into sentiments toward the vaccine and vaccination process. As illustrated through many of the quotes, hesitancy and vaccine uptake are not mutually exclusive: many of the patient narratives that characterize hesitancy were provided immediately following receipt of the vaccine and information provision. Sentiments toward medical intervention in general and sources of knowledge/information seemed to be underlying factors in hesitancy, as found in previous scholarship with migrants and racialized persons [19,29,30]. The study developed a typology of hesitancy (see Table 4) to map out the diverse reasons that shape refugees’ vaccination decisions. It is organized by hesitancy and vaccine motivating factors and captures themes respective to each factor in an exhaustive manner. While there may be some conceptual overlap in themes, the intent of the typology is to demonstrate the ways that hesitancy and motivation appeared in the data.

Hesitancy is a complex phenomenon that includes a range of categories, such as the refusal to be vaccinated, those who delay vaccination, and those who accept vaccination with some reservations. Participant perspectives on vaccination ranged from refusal to confidence, with the risk of negative health consequences being a major deciding factor for both groups. For others, hesitancy was presented as trust in one’s innate ability to fight the COVID-19 virus or a lack of confidence in vaccine effectiveness, while decisions to be vaccinated could be influenced by vaccine mandates or family pressure. Information and knowledge sources had the potential to validate or challenge the perspective of both vaccine hesitant and vaccine confident folks. For many, vaccination decisions were a negotiation between personal thoughts toward medical intervention, personal observation, family and community perspectives and opinions, news media, social media, direct public health guidance, and scientific evidence. A key recommendation from both patients and key informants regarding hesitancy was to ensure that ample, comprehensive information was reaching communities on an ongoing basis, from trusted sources such as personnel who were knowledgeable of their culture.

Also consistent with previous research was an emphasis on structural and systemic determinants of vaccine refusal and uptake [7,16,17,18]. Lack of first-language services, system complexity, and impersonal, clinical environments can all influence vaccination rates. Suggested solutions included tailored outreach plans, first-language information provision and service delivery, accessible, community-based vaccination sites with flexible booking options and hours, and partnership with non-governmental organizations to facilitate information provision, access, and tailored services to explicitly address immigrant hesitancy, stigma and risks associated with COVID-19 [36]. For example, the cohorts of Afghan refugees in processing hotels had most transportation barriers eliminated by the host immigrant service organization and clinic by providing services on site. They also had on-site international medical graduates to educate patients about the vaccine, help them navigate the system, and learn about supports in place for their specific needs.

Third, vaccine confidence and uptake were facilitated by various personal and external motivators, some of which overlapped with efforts of vaccination staff. In addition to hesitancy, there were many instances of patients eager to be vaccinated or to have family members become eligible for vaccines. Confidence was largely influenced by access to information, knowledge networks, and trust. Importantly, the intent to be vaccinated is not necessarily predictive of the decision to be vaccinated. Our study included some participants who, though vaccinated, would have preferred not to have been vaccinated or did not intend to receive further doses. This further supports the argument that hesitancy should be framed as part of an ongoing set of decision-making processes rather than a finite decision, as patients shared a number of these contradictions.

Lastly, a key learning from this study was that tailored approaches for refugees are potential opportunities to build trust and bridge clients to mainstream services in the future. Refugees are a unique group because they are some of the most vulnerable members of Canadian society. Given that they had likely experienced extreme circumstances prior to arrival in Canada, the vaccination roll-out at the early stage in their migration process to Canada became a significant site for engagement, trust building, and importantly, communicating the relationship between community and the health system. For refugees, this becomes a crucial entry point.

This study does not represent an exhaustive view of the experiences of refugees within Calgary area vaccination systems. Researchers primarily focused on recently arrived government-sponsored Afghan refugees and other Privately Sponsored Refugees due to the partnership with CCIS and AIMGA, as CCIS was the designated immigrant serving organization to host and process Afghan refugees in the area. AIMGA, a partner of CCIS, provided various cultural, medical, and language supports alongside CCIS. The partnership ultimately facilitated recruitment towards participants that were connected with those organizations and was used as a basis to reach a population that is difficult to access by drawing on previously established contacts and networks of trust. Furthermore, the study design was not structured to draw conclusions through statistical analysis. As with most qualitative research, this research presents insights on vaccination from a particular group of people in one geographic region, and may not reflect the diversity of experiences of refugee patients.

## 5. Conclusions

This study of refugee experiences with COVID-19 vaccinations in the Calgary area explored the experiences of refugee patient experiences within systems and how health systems adapted to meet the needs of refugees. By focusing on the experiences of a specific set of patients in the healthcare system, we gained insight into how multiple factors shape decisions to vaccinate, and how healthcare systems and partnerships can shape service delivery to adroitly address the needs of patients who face multiple intersecting vulnerabilities. In this case, it is argued that despite diverse individual, interpersonal, community, and societal factors that shape vaccination decisions and contribute to lower vaccine uptake, vaccine uptake for refugees can be promoted by fostering positive relationships and trust between vaccination personnel and refugee patients. As such, the health system can adapt to hesitancy and barriers to vaccination for vulnerable populations even in times of rapid change and massive upheaval, as shown by our study. To promote vaccine uptake in a context of hesitancy and barriers to vaccination for refugees requires a multipronged approach, which includes: (a) integrating culturally responsive healthcare delivery that meets refugees where they are; (b) barrier reduction measures, such as timely and credible information in first languages, on-site translation by medically trained personnel, transportation, on-site vaccinations, and extended hours of services; (c) trust building through engagement; (d) building educational and outreach partnerships with private groups, community organizations, and leaders.

## Figures and Tables

**Figure 1 vaccines-12-00177-f001:**
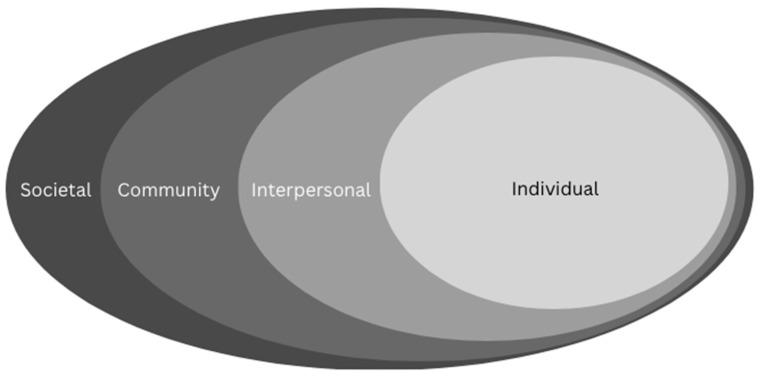
A social ecological model of health. Adapted from the Social-Ecological Model: A Framework for Prevention [35].

**Table 1 vaccines-12-00177-t001:** Breakdown of participant characteristics by position, language, and gender, for key informants, refugees, and sponsors in Calgary, 2021–2022 (N = 61).

Key Informant Breakdown by Position and Gender (n = 13)							
Position in Vaccination Systems						Gender (n)	
Medical Professionals (n)	Public Health Representatives (n)	Immigrant Serving Agency Staff (n)		Community Advocates (n)	International Medical Graduates and Other Staff (n)	Men	Women
2	2	5		2	2	7	6
Refugee and Sponsor Breakdown by Language and Gender (n = 48)							
Participant Type (n)	Language of Interview (n)					Gender (n)	
	Dari	Pashto	Arabic	Amharic	English	Men	Women
GAR (n = 39)	20	19				26	13
PSR (n = 6)			6			4	2
Sponsor (n = 3)			1	1	1	1	2

**Table 2 vaccines-12-00177-t002:** Accessibility and barrier factors, structural factors, and other determinants related to COVID-19 refugee vaccination discussed by participants in Calgary and surrounding areas, 2021–2022.

Vaccine Accessibility/Barriers	Structural Factors	Other Determinants
Appointment times	Vaccine eligibility	Legal status
Booking pathways (skill requirements, complexity, wait times)	Mandates or incentives	Time in Canada
Geography and transportation	Access to tailored models	Knowledge of English
Availability of reliable information	Public health information	Literacy levels
Access to first language information		Pre-migration experiences with health systems
Lack of cultural or faith accommodations		Level of Education
Lack of first language services		
Employer resistance to vaccinations		
English bureaucracy		

**Table 3 vaccines-12-00177-t003:** Verbatim quotes from GAR and Key informant participants regarding factors that impact decisions to vaccinate for COVID-19 in Calgary and surrounding areas, 2021–2022.

Factors	Responses
Individual, interpersonal, and community factors	
Concerns about side effects:	“Yes, for the first dose, I was very worried as the rumors were there, and in Kabul they were saying the vaccines are outdated and it may have lots of side effects in the future. I was worried that the side effects could affect me. But in the second dose I was not worried as it was well explained to me what the effects of the vaccine are”. (GAR participant 05)
Personal ‘disbelief’ in vaccine necessity or effectiveness and/or preference to avoid medical intervention:	“Personally, I do not believe in COVID vaccine myself and since I have not used lots of medicine since my childhood so I think my body should be able to fight for me”. (GAR participant 20)
Concerns about risks to subpopulations:	“Sometimes If there is a pregnant woman or any kids or children [that] are suffering from congenital disease or abnormalities, their parents, they have some concern”. (Key Informant participant 07)
Misinformation:	“I have a sister [who] has not visited in a long time just so she doesn’t have to take the vaccine (laughs)…She is the type of person that anything she watched on the news or social media she straight away believes. She was saying no, God forbid, this is bad for you, and all these sorts of stories”. (PSR participant 06)
Pre-migration experiences:	“The first two dose were done in Afghanistan… When the COVID vaccine came in, the first 6 months it was only the governmental people who were receptive. No other ordinary person was talking about the vaccine, and no one wanted to get the vaccine. But later everyone wanted to get the vaccine and I had to visit many clinics to be able to get my shot. So eventually after visiting several clinics, I went to a clinic for the vaccination in the morning and I only received at 3 pm, I had to wait for a long time as there were many people who came for vaccination. In Afghanistan the vaccine availability was very low. My third dose was done in Greece”. (GAR participant 02)
Fatigue, indifference, and booster-specific hesitancy:	“… As for my wife, she didn’t want to take the third shot, I tried to convince her to take the third shot, but she refused, she said two shots are enough”. (PSR participant 01)
Fear of adverse consequences:	“Sometimes If there is a pregnant woman or any kids or children [that] are suffering from congenital disease or abnormalities, their parents, they have some concern”. (Key Informant participant 07)
Information overload:	“It was overwhelming, we had some people sit at our desks and agonize over, you know, Moderna or Pfizer. For and you know, and I’m saying it’s Coke or Pepsi. Just you know, just take one right, and but they just like “well I read about this one and it looks like this one is 93% effective. But now we have a new variant, and what if it’s not as effective on the new variant” and you could see people kind of spinning with all this information right and fair enough. Because it was [difficult]. But yeah, we definitely found it was an overwhelming lot of information for a lot of people especially I would say people that have either a medical or a science background were almost overthinking it, too much, too much research”. (Key Informant participant 02)
Desire to protect others:	“I did not want to be contagious and transfer COVID to other people, so I went for my vaccination. It was only me who decided to go for vaccination”. (GAR participant 06)
Influence of family members:	“If you want me to tell you honestly. I don’t like vaccines. For me, I did not want to get it done. However my husband, you can say he decided for all of us. He said we have to take the vaccine, so we all took the vaccine”. (PSR participant 06)
Access to evidence-based information:	“Before my doses of COVID vaccination in Afghanistan and in Canada, they told me about the effects of vaccines. There were nurses both in Afghanistan and in here and I had enough time to ask them any questions. Their behaviour, it was very similar to those in here (Canada) and I think they did well. They made me aware of the effects of vaccine and helped me to be ready mentally for the vaccination. Everyone at the clinic was helpful but I think it was more nurses who were involved”. (GAR participant 20)
Secondary information sources and personal networks:	“Social Media, Facebook, TikTok, many stars, like the president, football players all went on and encouraged everyone to go for vaccination. In Greece, there were many nurses and there were some psychiatrists who gave us energy and encouragement to go for vaccination”. (GAR participant 07)
Societal factors	
Vaccine incentives and mandates:	“I had to get the vaccine as it was a necessity for my work and my travel, so I had to go for vaccination. If it was to myself, I probably may not go for vaccination. My family [also played] a role, and they were very important [as I wanted my] family not to be in any danger of not being vaccinated. It is for the formal things rather than my own choice, traveling to some places want you to have your vaccination”. (GAR participant 20)
Shifting public health information:	“[…] so many people took Sinopharm, so many people took Johnson and Johnson [prior to being with our clinic]. And the system just said no, this is not something legit or not something approved by our healthcare system. You have to take a full new series, either Moderna or you have to take Pfizer. In just like 2–3 weeks after that, they put on their website like this is all OK”. (Key Informant participant 01)
Accepted vaccines by public health agencies:	“My only suggestion is that during my stay as a refugee, I experienced the hard days. Many of my friends have had five doses of vaccine. Pakistan does not accept the vaccine that was given in Afghanistan. In Canada they do not accept the one that was given in Albania. I suggest that these countries become united, and they accept each other’s vaccine”. (GAR participant 05)

**Table 4 vaccines-12-00177-t004:** Typology of hesitancy and motivating factors.

Hesitancy Factors			Vaccine Motivating Factors
Refusal to be vaccinated	Delay vaccination	Accept vaccine but maintain sentiments of hesitancy	Vaccine confidence
Do not believe in vaccine necessity or effectiveness (may be all or specific vaccines)	Preference to wait for more research, more definitive/convincing research, and/or critical mass of vaccination	Mandates/unreasonable consequences	Fear of personal, family or broader social health consequences of not being vaccinated
Do not believe in medical intervention globally (medicine, vaccines) and/or belief in body’s inherent capacities	Waiting to see what happens in social networks/anecdotal evidence	Pressure from family members	Access to evidence-based information
Vaccine ingredients (not compatible with faith, allergies)	Undecided	Family members at high risk	Personal capacity and ability to navigate competing sources of information
Lack of confidence in vaccine effectiveness/potential side effects	Fear of negative side effects, influenced by knowledge sources and personal observation		Pressure from family members
Refusal of booster doses only (possible vaccine fatigue)	Inconvenience/competing priorities		

## Data Availability

The datasets analysed during the current study are not publicly available due to ethical considerations but are available from the corresponding author on reasonable request.

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
