# Peer review of "COVID-19 Vaccinations, Trust, and Vaccination Decisions within the Refugee Community of Calgary, Canada"

_vaccines, 2024, doi:10.3390/vaccines12020177_

Round 1

Reviewer 1 Report

Comments and Suggestions for Authors

Congratulations to authors for very well written manuscript and addressing very important issue. Although sample size was very small (N-61), hence findings cannot be generalized. It will be more beneficial if authors can quantify factors and determinants. 

thanks

Author Response

Thanks for this suggestion. While we agree that quantifying factors and determinants can be a good way for readers to visualize research and generate insights, due to the scope of the research, the data collection methods, and what we were able to collect we are unable to address this point outside of a basic demographic table at the beginning of Results (line 248). We have also included some general information on lines 236-247 detailing what information we were able to collect. Furthermore, with a sample of 61 participants, the variables available we did not conduct any statistical analysis.

In essence, we were able to confidently report on key informant position, gender for all participants, and the language of interviews. Although we started to collect demographic information from refugee (GARs, PSRs) and sponsor participants such as age and country of origin, we were unable to systematically collect this information due to multiple factors. These factors include working with intermediaries, patients not answering questions, and the information not being made available. Since this information was not reliable for the various groups, we decided to omit reporting on it.

Lastly, given that this research was qualitative in nature we did not set out to draw out any findings that were generalizable to the population, and instead set out to explore rich descriptions and to communicate insights for public health efforts based on existing health efforts aimed at refugees. As such, we did set up the study to collect data that could be quantified outside of what was outlined above. This was done in response to an identified need for qualitative scholarship on the experiences of refugees, which resulted in research questions that were qualitative in orientation and research methods that were aligned with the research questions. An addition has been made in the limitations section on lines 513-514 to reflect this.

Reviewer 2 Report

Comments and Suggestions for Authors

The manuscript seeks to carry out an exploratory qualitative approach to immigrants/refugees during and after the pandemic process in a city in Canada for health policy recommendations. The basis of the study is the information and experience provided by refugees.

The manuscript needs to be clearer, with repetitive sections and structurally poorly presented. Therefore, we do not recommend its publication in this format.

Major points

1) It Is necessary to complete a description of the Institutions involved.

2) The methodology is described continuously without revealing what is being analyzed at each stage.

3) Table 1 can be presented as supplementary, as it is not data from the authors,

4) We missed using statistics to draw factual conclusions despite the manuscript's "qualitative" focus.

5) the MM and results must present sections covering each subject's titles.

6) The conclusions could be clearer, and the manuscript needs to clarify what was concluded.

7) Authors' contributions must comply with the journal's rules, presenting the authors' names in abbreviated form,

8) The description of references must also comply with the journal's standards

Comments on the Quality of English Language

It is okay; only minor editing of English is necessary.

Author Response

The manuscript seeks to carry out an exploratory qualitative approach to immigrants/refugees during and after the pandemic process in a city in Canada for health policy recommendations. The basis of the study is the information and experience provided by refugees.

The manuscript needs to be clearer, with repetitive sections and structurally poorly presented. Therefore, we do not recommend its publication in this format.

Response: Thank you for your feedback on our manuscript. We've addressed concerns through addressing the major points. With these changes, we believe the paper is now clearer and suitable for publication. We hope you reconsider our submission.

Major points

1) It Is necessary to complete a description of the Institutions involved.

Response: Thank you for pointing out a need for a description of the institutions involved. A paragraph has been added at the beginning of the Methods section with descriptions of the institutions involved on lines 117-136.

2) The methodology is described continuously without revealing what is being analyzed at each stage.

 Response: Thank you for pointing out issues with the methods section. We have revised the latter part of the methods section to make is clearer to the reader how the research team conducted the analysis. The reworked section can be found on lines 217-232. This includes a step-by-step description of the thematic analysis. The methods section has also been reorganized based on reviewer comments.

3) Table 1 can be presented as supplementary, as it is not data from the authors,

 Response: Thank you for the suggestion regarding Table 1. The table has been included as supplementary material (Table S1: Determinants of undervaccination among migrant populations) and referenced accordingly in lines 59-60. In-text citations of the referenced materials in the table have also been made in accordance with the journal guidelines on lines 56-60.

4) We missed using statistics to draw factual conclusions despite the manuscript's "qualitative" focus.

Response: Thanks for this suggestion. While we agree that quantifying factors and determinants can be a good way for readers to visualize research and generate insights, due to the scope of the research, the data collection methods, and what we were able to collect we are unable to address this point outside of a basic demographic table at the beginning of Results (line 248). We have also included some general information on lines 236-247 detailing what information we were able to collect. Furthermore, with a sample of 61 participants, the variables available we did not conduct any statistical analysis.

In essence, we were able to confidently report on key informant position, gender for all participants, and the language of interviews. Although we started to collect demographic information from refugee (GARs, PSRs) and sponsor participants such as age and country of origin, we were unable to systematically collect this information due to multiple factors. These factors include working with intermediaries, patients not answering questions, and the information not being made available. Since this information was not reliable for the various groups, we decided to omit reporting on it.

Lastly, given that this research was qualitative in nature we did not set out to draw out any findings that were generalizable to the population, and instead set out to explore rich descriptions and to communicate insights for public health efforts based on existing health efforts aimed at refugees. As such, we did set up the study to collect data that could be quantified outside of what was outlined above. This was done in response to an identified need for qualitative scholarship on the experiences of refugees, which resulted in research questions that were qualitative in orientation and research methods that were aligned with the research questions. An addition has been made in the limitations section on lines 513-514 to reflect this.

5) the MM and results must present sections covering each subject's titles.       

Response: Thank you for highlighting a need for a different organization of the methods section. The methods section has been revised with some details added (in accordance with some on the points above) which are highlighted in yellow on lines 117-136, 171-186. Furthermore, the methods section has been organized under specific headers (Research Design, Sampling and Recruitment, Data Collection, Data Analysis), with some of the content moved around to fit with this framework.

 6) The conclusions could be clearer, and the manuscript needs to clarify what was concluded.

Response: Thank you for this suggestion regarding the clarity of study conclusions. The Conclusion section has been revised, with new content added on lines 519-528 to make it clearer to the reader what was concluded.

 7) Authors' contributions must comply with the journal's rules, presenting the authors' names in abbreviated form,

Response: Thank you for pointing this out. The author’s names now appear in abbreviated form on lines 539-545.

 8) The description of references must also comply with the journal's standards

Response: Thank you for pointing out that the references need to be formatted to the specific journal format. The references and in-text citations are now in the format of the journal. References start on line 562.

Reviewer 3 Report

Comments and Suggestions for Authors

I have no major concerns with this paper and appreciate the opportunity to review it.  It is solid, and while it doesn’t add anything especially new in terms of recommendations or conclusions, it is consistent with the pre-existing literature, and does add to the growing literature in this area by focusing on a particular demographic—refugees in Calgary.

My few (minor comments):

  1. The term ‘racialized’ introduced on line 49 and used throughout is problematic. I think I know what it means, but it raises some red flags.  It should at least be carefully defined if not changed to something like ‘marginalized’. . .

  2. What is the larger study referred to on line 116?  Has it been published on previously?  Is it relevant here to know?

  3. Line 153 has grammatical/tense problems.

  4. Reword on line 157 and throughout: ‘key learnings’.  What does this mean? Key outcomes?  Unclear

  5. Line 199.  What is ‘the research’ referred to here?

  6. I like the model, Figure 1, etc, although it’s a little simplistic, it works and fits well with the data obtained and vice versa.

  7. The use of the word ‘several’ a few times on page 6 is vague.  Give a number or percentage.

  8. Line 266.  Per say should be per se

  9. Line 301 and in other places following: ‘structural’ is vague.  Does this refer to infrastructure or what exactly?  Please define more carefully.

  10. Line 305.  ‘Correlated’, unless i’m missing some data or point here, should be ‘ascribed’ or something similar, instead of 'correlated' which suggests much.

  11. Table 3 is effective.  ‘Verbatim quote’ seems redundant.  Maybe just ‘Responses’?

  12. Line 349: not sure what ‘it has also actively provided insight’ means.

  13. Line 352-3: again problems with words in data interpretation.  I think it should say, instead of ‘Refugees demonstrated. . .’, rather ‘We provided evidence to support . . .

  14. I like Table 4, but why is there no ‘Vaccine Motivating Factors’ listed for the last row?

Author Response

My few (minor comments):

  1. The term ‘racialized’ introduced on line 49 and used throughout is problematic. I think I know what it means, but it raises some red flags.  It should at least be carefully defined if not changed to something like ‘marginalized’. . .

Response: Thank you for pointing out the need for a core term definition. We have changed the word to “marginalized”.

2. What is the larger study referred to on line 116?  Has it been published on previously?  Is it relevant here to know?

Response: Thank you for the questions regarding line 116. While this research paper is an outcome of a broader research study which resulted in another paper on COVID-19 models of vaccination for refugees and newcomers in Calgary (currently under a second round of peer review\), we felt it was not necessary to state this. We have decided to omit the words “Embedded within a broader research study on models of vaccination and patient experiences,” and begins the sentence with “This paper…” on line 140.

3. Line 153 has grammatical/tense problems.

Response: Thank you for identifying grammatical problems. We have reviewed the original sentence on line 153 and it now appears on lines 193-195, in a correct format. We made changes to make the sentence clearer to the reader.

4. Reword on line 157 and throughout: ‘key learnings’.  What does this mean? Key outcomes?  Unclear

Response: Thank you for pointing this out. We have decided to change the word learnings to lessons, to reflect that we asked key informants about lessons and recommendations based on their experience with vaccinations, as this is of interest to persons in public health roles. For example, lessons could be about changing vaccination practices to suit refugee needs, and that working with refugees helped build the capacity of vaccination personnel to work in intercultural settings, to name some. The change has been made to line 198.

5. Line 199.  What is ‘the research’ referred to here?

Response: Thank you for pointing out issues with language. We have changed the word ‘the’ to ‘this’ to be explicit. This is reflected on line 257.

6. I like the model, Figure 1, etc, although it’s a little simplistic, it works and fits well with the data obtained and vice versa.

Response: Thank you for the insight regarding figure one. Based on this comment, we have taken no action.

7. The use of the word ‘several’ a few times on page 6 is vague.  Give a number or percentage.

Response: Thank you for this out. We have revised the manuscript accordingly where several is stated in that section and given a specific number of refugee responses that fit with the particular statement on p.7, lines 277 and 286, in parentheses.

8. Line 266.  Per say should be per se

Response: Thank you for pointing out issues with spelling. This has been addressed and is reflected on line 324.

9. Line 301 and in other places following: ‘structural’ is vague.  Does this refer to infrastructure or what exactly?  Please define more carefully.

Response: Thank you for pointing out a need for a definition. We have replaced the word structural with societal on line 359 for consistency and added applied definitions for societal level factors and structural factors on lines 359-364

10. Line 305.  ‘Correlated’, unless i’m missing some data or point here, should be ‘ascribed’ or something similar, instead of 'correlated' which suggests much.

Response: Thank you for pointing out this issue around language. We have replaced the word correlated with ascribed, as this accurately reflects our interpretation of this phenomenon. It is now on line 368.

11. Table 3 is effective.  ‘Verbatim quote’ seems redundant.  Maybe just ‘Responses’?

Response: Thank you for the suggestion. We have changed Verbatim Quotes with “Responses” in Table 3.

12. Line 349: not sure what ‘it has also actively provided insight’ means.

Response: Thank you for pointing out this phrase. We have removed “actively” from the sentence on line 412.

13. Line 352-3: again problems with words in data interpretation.  I think it should say, instead of ‘Refugees demonstrated. . .’, rather ‘We provided evidence to support . . .

Response: Thank you for pointing out these subtleties in language. We have revised the sentence in accordance with your suggestion on p. 415-416.

14. I like Table 4, but why is there no ‘Vaccine Motivating Factors’ listed for the last row?

Response: Thank you for pointing this out. The table is an exhaustive list of types of hesitancy, and types of motivating factors that challenge hesitancy, based on the data. Given that it was exhaustive and that we did not come across any more types of motivating factors, we did not have a vaccine motivating factor in the last row. Furthermore, we acknowledge that the layout can cause confusion, and as such we have have added a short preamble to the reader on lines 455-458.

Round 2

Reviewer 2 Report

Comments and Suggestions for Authors

The manuscript covered a profound construction remodeling; however, the text still needs revision. Many words and sentence constructions appear sequentially (e.g., lines 277/286). We suggest that an analysis be carried out throughout the text.

Although the work informs that only qualitative analysis is conducted, too many individuals are consulted to provide a general opinion and solid conclusions. In this aspect, the robustness of the work loses its importance due to the impossibility of applying statistical methods.

Comments on the Quality of English Language

English is fine, only minor editing words are required

Author Response

Response: We have read the reviewer’s latest response and have interpreted this as two points to consider.

1) We appreciate the push towards a rigorous level of writing. The text was reviewed with minor edits made throughout and repetitions were removed throughout the manuscript. The Results section was also modified more significantly, and we hope issues around word and sentence sequences were addressed through the removal of repetitions and by changing how the contents of the themes were presented.

2) Regarding the second point, we have interpreted this as a comment regarding the methodological limitation of qualitative methods and/or studies that solely use qualitative methods. To address concerns surrounding our use of qualitative data and how we came to these conclusions, we have included text on lines 140-148 that references the need for this type of data for vaccine interventions, and ways that we addressed issues of qualitative validity and reliability through our research design. We hope that this addresses any concerns.

Round 3

Reviewer 2 Report

Comments and Suggestions for Authors

The new modifications introduced and justifications presented are convincing. We can now consider the manuscript ready to be published.